# Spatial and Temporal Variations in PM_10_ Concentrations between 2010–2017 in South Africa

**DOI:** 10.3390/ijerph182413348

**Published:** 2021-12-18

**Authors:** Oluwaseyi Olalekan Arowosegbe, Martin Röösli, Temitope Christina Adebayo-Ojo, Mohammed Aqiel Dalvie, Kees de Hoogh

**Affiliations:** 1Department of Epidemiology and Public Health, Swiss Tropical and Public Health Institute, Socinstrasse 57, CH-4002 Basel, Switzerland; martin.roosli@swisstph.ch (M.R.); temitope.adebayo@swisstph.ch (T.C.A.-O.); c.dehoogh@swisstph.ch (K.d.H.); 2Faculty of Science, University of Basel, CH-4003 Basel, Switzerland; 3Centre for Environmental and Occupational Health Research, School of Public Health and Family Medicine, University of Cape Town, Rondebosch, Cape Town 7700, South Africa; aqiel.dalvie@uct.ac.za

**Keywords:** particulate matter pollution, PM_10_, South Africa, spatial, temporal

## Abstract

Particulate matter less than or equal to 10 μm in aerodynamic diameter (PM_10_ µg/m^3^) is a priority air pollutant and one of the most widely monitored ambient air pollutants in South Africa. This study analyzed PM_10_ from monitoring 44 sites across four provinces of South Africa (Gauteng, Mpumalanga, Western Cape and KwaZulu-Natal) and aimed to present spatial and temporal variation in the PM_10_ concentration across the provinces. In addition, potential influencing factors of PM_10_ variations around the three site categories (Residential, Industrial and Traffic) were explored. The spatial trend in daily PM_10_ concentration variation shows PM_10_ concentration can be 5.7 times higher than the revised 2021 World Health Organization annual PM_10_ air quality guideline of 15 µg/m^3^ in Gauteng province during the winter season. Temporally, the highest weekly PM_10_ concentrations of 51.4 µg/m^3^, 46.8 µg/m^3^, 29.1 µg/m^3^ and 25.1 µg/m^3^ at Gauteng, Mpumalanga, KwaZulu-Natal and Western Cape Province were recorded during the weekdays. The study results suggest a decrease in the change of annual PM_10_ levels at sites in Gauteng and Mpumalanga Provinces. An increased change in annual PM_10_ levels was reported at most sites in Western Cape and KwaZulu-Natal.

## 1. Introduction

The levels of air pollution in sub-Saharan Africa (SSA) have remained high compared to other regions of the world that have witnessed notable improvements [1]. The deteriorating trend of air quality in SSA countries, such as South Africa, has been linked to rapid urbanization, industrialization and the resultant increase in population. South Africa relies significantly on fossil fuel for both industrial and domestic activities—over 80% of power generation is from fossil fuel. Other important sources of air pollution emission in South Africa include bush burning, land-fills, dust from construction sites and wind-blown dust from open land [2,3]. Exposure to ambient air pollution accounted for over four million deaths globally in 2019 [4]. Particulate matter less than or equal to 10 μm in aerodynamic diameter (PM_10_ µg/m^3^) is one of the most important pollutants of public health interest that is monitored in South Africa [5]. The revised 2015 National Air Quality standard of daily limit of 75 µg/m^3^ and annual limit of 40 µg/m^3^ are less stringent than the World’s Health Organization’s limit of 45 µg/m^3^ and annual limit of 15 µg/m^3^ [6,7].

The levels of PM_10_ concentration can vary in space and time due to distinct meteorological conditions and anthropogenic sources, such as vehicular, domestic and industrial emissions, between the different provinces in South Africa [8,9]. Several air quality management policies and strategies have been introduced to address the worsening air quality in South Africa [6]. These include the identification and control of priority pollutants, the promulgation of regulations to reduce emissions from industries and the classification of air pollution prone areas as priority areas for efficient management of limited air quality management resources [6,10,11]. To this end, air quality management and monitoring resources are concentrated in four air pollution priority areas, including the Highveld, the Vaal triangle, the South Durban Basin and Waterberg, located in four different provinces (Gauteng, Mpumalanga, Western Cape and KwaZulu-Natal) of South Africa. These four areas were prioritized due to the propensity of the observed or outlook of air quality in these areas to exceed the national air quality standards [6,10,12]. Only a few previous studies of air pollutants have examined the spatial and temporal trends of PM_10_ from sites in these areas and SSA [10,12,13,14,15]. This is because of limited measurement data to explore the long-term spatial and temporal patterns of PM_10_ in these areas. A couple of studies have assessed the trend in PM_10_ mostly in air pollution priority areas of Gauteng and Mpumalanga province of South Africa [10,12,13,15,16]. In addition, Onyango et al. described the spatial and temporal variation in PM_10_ concentrations at three sites in Uganda [14].

Our previous study described the quality of ground-level PM_10_ measurements in four provinces of South Africa, Gauteng, Mpumalanga, Western Cape and KwaZulu-Natal, for the years 2010–2017 [17]. The earlier study explored methods to bridge the gap in daily PM_10_ data by imputing missing daily PM_10_ for some sites in these provinces for the study period. This study intends to build on the PM_10_ exposure data from the earlier study to characterize daily PM_10_ spatially and temporally for four provinces of South Africa. To investigate the pattern of change in PM_10_, we assessed the change in annual PM_10_ average across the sites in these areas for the years 2010–2017. Additionally, we explored the characterization of potential influencing factors of PM_10_ emission around the sites. An improved understanding of the pattern of PM_10_ concentration between the four provinces can play a significant role in informing mitigation actions toward addressing the threat posed by air pollution, especially in low- and middle-income countries, such as South Africa, with limited ground-monitored data.

## 2. Materials and Methods

In this study, PM_10_ measurements from 44 monitoring sites across four provinces (Gauteng, Mpumalanga, Western Cape and KwaZulu-Natal) of South Africa were included. Hourly PM_10_ from the South African Air Quality Information System (SAAQIS). SAAQIS can be reached via their website (https://saaqis.environment.gov.za/, accessed on 22 October 2018). For our study, we selected, for each year between 2010 and 2017, all sites with more than or equal to 70% of total daily measurement data available during a year [17]. Missing data were imputed using a random forest machine learning method, including spatiotemporal predictors, like meteorological, land use and source-related variables, as described in detail in our previous paper [17]. The combined observed and imputed data were used for this study analysis. The distribution of the sites across the provinces differs substantially (Figure 1). The Vaal triangle airshed Priority Area monitoring network and the Highveld Priority Area air quality-monitoring network that cut across Mpumalanga and Gauteng Provinces were the earliest networks established to monitor ambient air quality in South Africa. The South Africa Weather Service classifies the majority of the sites (21) as industrial sites, 18 as residential sites and 5 as traffic sites. An overview of the state of annual PM_10_ availability is presented in Appendix A. Thirty-two of the forty-four sites (73%) have more than a year of PM_10_ measurement data.

To evaluate the potential influencing factors of PM_10_ around these monitoring sites, we explored multiple buffers (100, 300, 500, 1000, 10,000 m) of land use categories (Residential and Industrial), population density and road density around the monitoring sites. South Africa’s road network was obtained from OpenStreetMap (OSM) and the sum of road length was calculated for two categories: (1) major roads defined as roads of OSM types of primary, secondary and tertiary roads and (2) all roads defined as roads of OSM types of residential, service, motorway and trunk. Population density was obtained from the Socioeconomic data and Application Center (SEDAC) dataset. Land use was classified based on the 2018 South Africa National Land cover dataset categories.

To evaluate changes in annual average PM_10_ concentrations for 2010–2017, we applied two formulas. For sites with two consecutive years with average PM_10_ data, the change was calculated by applying the formula:(1)$$△\;=\left(\frac{Cx}{Cy}-1\right)*100$$
where Δ is the change, *Cx* is the annual mean PM_10_ concentration in the current year and *Cy* is the annual mean PM_10_ concentration in the previous year.

For sites with missing data between successive years, the change in average PM_10_ for a year with average PM_10_ data was calculated by applying:(2)$$△\;=(\frac{\frac{Cx}{Cy}}{y-x}-1)*100$$
where Δ is the change, *Cx* is the annual mean PM_10_ concentration in current year *y*, *Cy* is the annual mean PM_10_ concentration in the next previous year (year *x*) with an annual mean PM_10_ concentration and *y* − *x* is the number of year(s) between available measurements.

We also calculated annual changes of PM_10_ levels for the 32 sites with more than a year of PM_10_ sites using a linear regression analysis.

## 3. Results

### 3.1. Characterization of Sites

Figure 2 presents the level of variation in potential land use, road density and population variables that can provide information about the likely prominent influencing factors of PM_10_ around the site types as designated by the South Africa Weather Service. Buffers of different sizes ranging from 100, 300, 500, 1000, and 10,000 m radii around the sites were considered. Figure 3 presents the analysis for a 300-m buffer. The other buffers sizes did not show substantially different patterns in their distributions. Generally, the distribution of calculated land use, road density and population within a 300 m buffer are in agreement with the monitoring site classification, although industrial land use was actually lower around industrial sites compared to the other two site types. Residential land use and population density were highest for residential classified sites. Major road density within a 300 m buffer was highest for monitoring sites classified as traffic sites.

### 3.2. Annual Change in Site’s Average PM_10_ µg/m^3^ Concentration over the Study Period

Table 1 shows how the levels of PM_10_ change across the sites for the years 2010–2017. In Gauteng province, the average change in annual PM_10_ concentration decreased in 5 of 8 sites (63%) (Table 1). In Mpumalanga province, the average change in annual PM_10_ concentration decreased in 9 of 12 sites (75%). The average change in annual PM_10_ concentration decreased in only 3 of 7 (43%) Western Cape Province sites. Similarly, a decrease in the average change in annual PM_10_ concentration was observed in only 2 of 5 (40%) KwaZulu-Natal province sites.

### 3.3. Monthly Differences in Daily PM_10_

A summary of monthly mean PM_10_ concentrations across all sites per province and across the years 2010–2017 is presented in Figure 3. The pattern in the levels of PM_10_ across the four provinces (Gauteng, KwaZulu-Natal, Mpumalanga and Western Cape) of South Africa suggests seasonal variation in monthly PM_10_ levels. The monthly PM_10_ levels show a seasonal pattern across the provinces and are more prominent in Gauteng and Mpumalanga provinces. The monthly mean PM_10_ levels were highest in Gauteng Province and lowest in Western Cape Province. In Gauteng, the lowest monthly mean PM_10_ concentrations were recorded during the summer months (December–February), ranging from a monthly mean of 15.51 µg/m^3^ recorded in December 2017 to 51.92 µg/m^3^ recorded in February 2012. The monthly mean PM_10_ peaked during the winter months, ranging from 35.29 µg/m^3^ recorded in July 2016 to 88.46 µg/m^3^ recorded in July 2011. The highest mean PM_10_ recorded during the winter months is about 5.7 times higher than the revised 2021 WHO annual PM_10_ air quality guideline of 15 µg/m^3^. In Western Cape Province, the lowest monthly mean during the summer months ranged from 15.53 µg/m^3^ recorded in December 2010 to 34.17 µg/m^3^ in February 2017. The monthly mean PM_10_ during the winter months ranged from 18.69 µg/m^3^ recorded in August 2012 to 34.98 µg/m^3^ recorded in June 2017. In general, all provinces recorded peak PM_10_ levels during the winter months in South Africa between June and August.

### 3.4. Week Day Differences in Daily PM_10_

Figure 4 summarizes daily mean PM_10_ per province for the eight-year study period. Generally, marginal differences were found between different days of the week between 2010 and 2017. Average daily PM_10_ concentrations during the weekdays are slightly higher than during weekends in all four provinces. The highest PM_10_ concentrations of 51.4 µg/m^3^, 46.8 µg/m^3^, 29.1 µg/m^3^ and 25.1 µg/m^3^ at Gauteng, Mpumalanga, KwaZulu-Natal and Western Cape Province were recorded during the weekdays. Statistically significant differences in mean PM_10_ concentrations were observed between weekdays and weekends (F = 14.57 and value = 0.0009) and by province (F = 380.11 and *p* value =< 0.0001). The Pairwise Tukey’s test comparisons suggest the difference between weekdays and weekends mean PM_10_ concentrations was statistically significant in all pairs of provinces but between KwaZulu-Natal and Western Cape (*p* value = 0.14).

### 3.5. Spatial Variation in PM_10_

A summary of descriptive statistics is presented in Table 2. There are 20 industrial sites, 18 residential sites and 5 traffic sites included in this analysis. These traffic sites are located in the three provinces of Gauteng (1 site), Western Cape (2 sites), KwaZulu-Natal (2 sites). The results from Table 2 show that the levels of PM_10_ concentration level is highest in Gauteng and for all provinces PM_10_ concentration is highest at the residential sites compared to industrial and traffic sites. In Gauteng, the concentration at one traffic site was similar to the concentration at the residential sites and substantially higher than the levels at the industrial sites.

The levels of PM_10_ concentration in Mpumalanga province are also high (Table 2). The industrial sites in Mpumalanga recorded the highest levels of PM_10_ compared to the industrial sites in other provinces. The PM_10_ concentration levels at both industrial and residential sites in Mpumalanga are comparable. The levels of PM_10_ concentration in Western Cape are the lowest compared to other provinces. Residential sites recorded the highest level of PM_10_ concentration in Western Cape. However, there are no substantial differences in the levels of PM_10_ concentration across the site types. Similarly, sites from KwaZulu-Natal province also recorded relatively low levels of PM_10_ concentration compared to Gauteng and Mpumalanga provinces. The levels at residential sites are also marginally higher than the levels at traffic sites (Table 2).

### 3.6. Attainment of PM_10_ Standards

Table 3 shows the percentage of days that PM_10_ concentration daily limits were exceeded using WHO and South Africa’s National Air Quality Standard (NAAQS) for the sites in the four provinces from 2010–2017. Gauteng province reported the highest proportion of days exceeding the daily limits of WHO and NAAQS standards, with about 38% of the days exceeding the WHO daily standard and around 17% of days exceeding the NAAQS daily standard. In contrast, Western Cape Province reported the lowest percentage of days exceeding both WHO and NAAQS PM_10_ air quality standards (3% and 0.09%, respectively) between years 2010–2017 (Table 3).

## 4. Discussion

This study adds to existing evidence on levels of PM_10_ in South Africa. The eight-year trend in PM_10_ level suggest that PM_10_ is still high in the earliest high pollution designated priority areas around Gauteng and Mpumalanga provinces. However, there is evidence of decreasing PM_10_ levels at most sites in both Gauteng and Mpumalanga Provinces. While the level of PM_10_ of most sites in KwaZulu-Natal and Western Cape Provinces suggest an increase in PM_10_ levels during the study period. The presented analysis identified trends in ambient PM_10_ concentrations in four South African Provinces for the years 2010–2017.

### 4.1. Spatial and Temporal Trends in Daily PM_10_

The Vaal Triangle Airshed Priority Area (VTAPA) and Highveld Priority Area around Gauteng and Mpumalanga Provinces were the first areas designated as air pollution priority areas in South Africa due to the observed or expected level of air pollution in these areas [2,10]. Both provinces share similar emissions profiles; they are home to the majority of coal-powered plants, coal mining, gold mining, mine tailing, petrochemical and ferroalloy industries in South Africa. To address the level of air pollution in these areas, air quality management plans were developed to guide actions towards improving the air quality in these priority areas. Some of the actions implemented to reduce the air pollution in these areas include the closure of a high polluting industry, large-scale domestic electrification program in these areas to reduce domestic emissions [2,10]. Thus, the decrease in the levels of PM_10_ reported in this study at most sites around these areas in Mpumalanga and Gauteng Provinces could be because of the changing emission profiles in these priority areas due to these mitigation actions. Despite the lower levels of PM_10_ reported in KwaZulu-Natal and Western Cape Provinces compared to Gauteng and Mpumalanga Provinces, the average change in annual PM_10_ increased at most sites in these Provinces. This trend signals a deteriorating air quality in these areas that is likely due to changes in emissions profiles in these areas. The Southern Basin Industrial areas in KwaZulu-Natal have been identified as air pollution hotspots due to the high density of industrial activities in this area [18]. There are also concerns about the air quality in Western Cape provinces, especially around the increasing informal settings in Western Cape Province [19].

The monthly variation in PM_10_ across the provinces during the study period shows that PM_10_ concentrations are highest during the winter months between June and August. This is consistent with results of a study in Gauteng assessing the characteristics of ground-monitored PM_2.5_ and PM_10_ between years 2010 and 2014 [13] and a study conducted in eMbalenhle—a low socio-economic in Mpumalanga province [15]. Similarly, the marginal seasonal difference in PM_10_ reported in Western Cape Province follows the pattern reported in a Western Cape study that reported the seasonal difference in PM_10_ in 2016 using data from one monitoring site [16]. A Ugandan study, however, reported higher PM_10_ concentration during the dry seasons compared to wet seasons [14]. The difference in seasonal weather patterns and sources of PM_10_ between South Africa and Ugandan could explain the seasonal difference in PM_10_ concentration. The cold season in most of South Africa’s provinces is characterized by cold weather and an increase in solid biomass use as a source of energy. The reliance of South African’s on solid biomass as a source of energy for cooking and heating system during the winter has been reported in other studies [2,20,21,22]. Residential fuel consumption in South Africa includes kerosene, residential fuel oil, LPG, sub-bituminous coal, wood/wood waste, other primary solid biomass and charcoal. Overall, residential fuel consumption dropped from 2010 to 2017, but domestic coal consumption increased slightly [22]. This study also highlights the fact that domestic sources of PM_10_ contribute substantially to the variability of PM_10_ in South Africa.

The trend in weekday PM_10_ level follows a similar pattern across the four provinces. PM_10_ concentration increased through the weekdays, reaching its peak between Wednesday and Friday. This study suggests that there is a difference between PM_10_ levels between weekends and weekdays, with lower PM_10_ levels reported during the weekends. Although there are only four traffic sites in this analysis, the decreased level in traffic-related activities during the weekend might be responsible for the observed lower PM_10_ levels during the weekend.

### 4.2. PM_10_ Level across Site Types

The trends in PM_10_ across the primary environment types in South Africa used for classifying the PM_10_ monitoring sites by the South Africa Weather Service are the Industrial, Residential and Traffic areas. The majority of the monitoring sites included in this analysis are industrial and residential sites. Table 2 shows that average PM_10_ levels were highest in residential sites compared to other categories of sites during the study period in all four provinces. This result is not unusual; similar results were reported in Gauteng areas of South Africa [13]. Our result also show that PM_10_ levels are generally higher at residential sites in the other three provinces. A possible explanation for the high levels of PM_10_ concentration levels at residential areas across the provinces is the high level of domestic burning in residential areas in South Africa. Previous studies have highlighted domestic emissions as the predominant source of particulate matter in South Africa [2,13]. It has also been argued that because the industrial emissions are released into a stable atmosphere in stacks above the generally shallow boundary layer height in South Africa could have affected the dispersion of the emissions to the ground level [2,13]. We also explored the variation in residential and industrial land use and road and population density around the monitoring sites. The residential radii have the highest level of variation from multiple influencing factors of PM_10_ emission. The high variability in the multiple sources of PM_10_ emission suggests that the high density of PM_10_ emissions around residential areas could explain the highest concentration of PM_10_ recorded in residential sites in our analysis. The high level of PM_10_ concentration and high variability of potential PM_10_ influencing factors around residential areas have implications on the population’s health outcomes [23].

### 4.3. Strengths and Limitations

There are some limitations in this study worth nothing. First, the pattern of missingness of PM_10_ exposure data during the study period poses a challenge to understanding the time-series trend in PM_10_ exposure data across the sites. The results presented are for four out of nine provinces in South Africa. Thus, these results cannot be extrapolated beyond the provinces that contributed data to our analysis. In addition, the representativeness of the site types is also a limitation of this study; the majority of the sites included in this study are industrial and residential sites. To address the challenge of missing daily PM_10_ data, this study combined observed and imputed PM_10_ exposure data from 44 monitoring sites across four provinces in South Africa to investigate the trends in PM_10_ concentrations. Despite the limitations, our results provide some insights on trends of PM_10_ concentrations in the four provinces during the study period.

## 5. Conclusions

It has been over a decade since the promulgation of South Africa’s National Environmental Management Air Quality Act in 2004. There have been concerns over the progress made so far [11]. We found that PM_10_ levels are higher than the WHO limits standard across the four provinces. The provincial differences in PM_10_ concentration show that PM_10_ levels are higher around air pollution priority areas, while the temporal variability of PM_10_ suggest that emissions during the winter months contribute markedly to the high level of PM_10_ recorded during the winter seasons.

An interesting result for future epidemiological studies in South Africa is the high level of PM_10_ and high variability of potential influencing factors of PM_10_ emission around where people live and work. Taken together, these results have implications for addressing the trends of PM_10_ pollution in South Africa.

## Figures and Tables

**Figure 1 ijerph-18-13348-f001:**
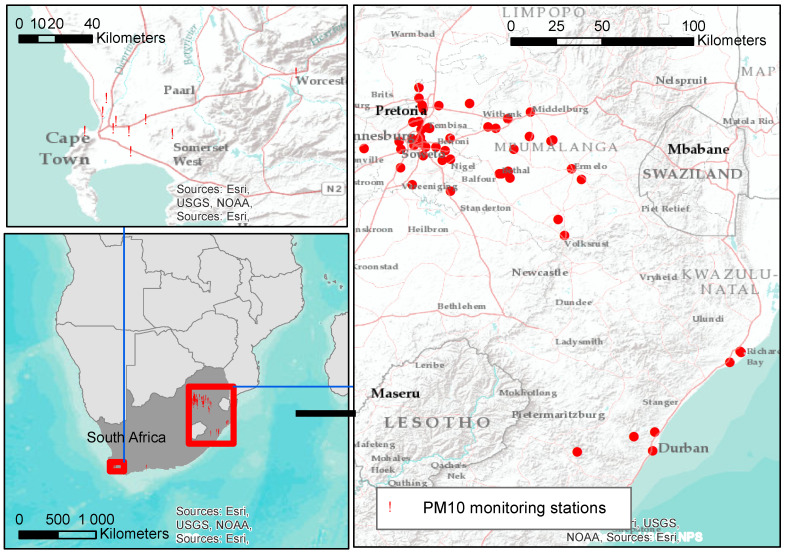
The spatial distribution of particulate matter (PM_10_) monitoring stations included in this paper across the four provinces of South Africa operating at some point during 2010–2017.

**Figure 2 ijerph-18-13348-f002:**
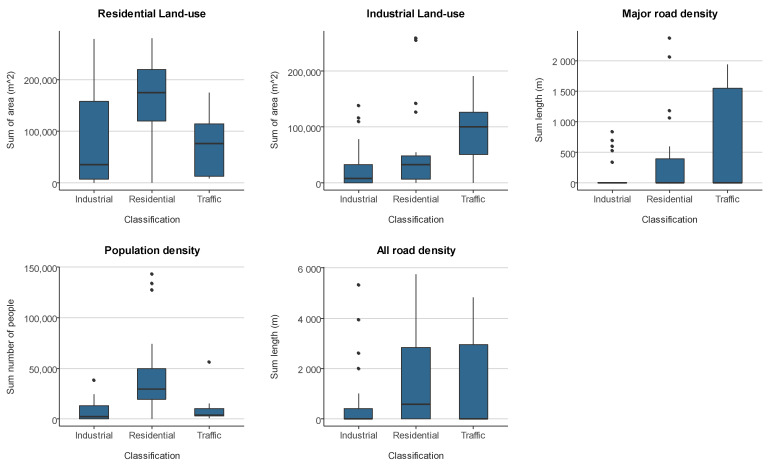
Distribution of indicators of PM_10_ emissions; land use (sum of area, m^2^), road density (sum length, m) and population (sum number of people) within a 300 m buffer across the three site classifications.

**Figure 3 ijerph-18-13348-f003:**
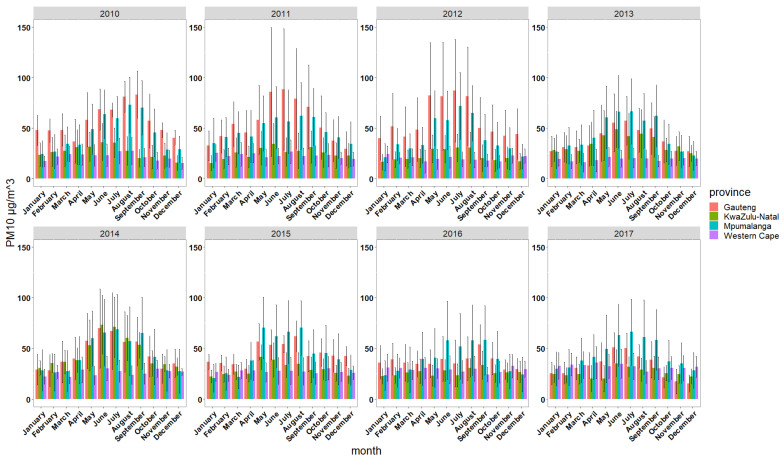
Variations in mean PM_10_ µg/m^3^ concentration across the provinces and months of the year. Error bars represent one standard deviation of the mean.

**Figure 4 ijerph-18-13348-f004:**
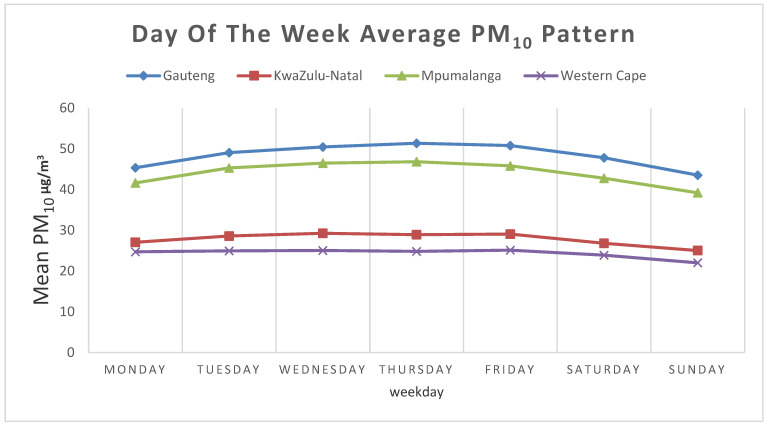
Weekdays variation in average daily PM_10_ µg/m^3^ concentration across the provinces.

**Table 1 ijerph-18-13348-t001:** Levels and changes in annual PM_10_ µg/m^3^ concentrations across sites for the years 2010–2017. The first entry per site shows the annual PM_10_ concentration in µg/m³. Subsequent entries depict the percentage changes compared to the previous entry. The last column shows the annual changes in µg/m³ per year assuming a linear trend between the first and last available measurement per site.

Province	Site	2010	2011	2012	2013	2014	2015	2016	2017	Annual Change in PM_10_
		Percentage increase/decrease								
Gauteng	Bodibeng			57.64	+2.54					+1.46
	Booysen			57.88		+7.42				+4.30
	Ekandustria				30.40	+70.51				+21.4
	Elandsfontein							29.45	−1.64	−0.48
	Leandra		22.14	−28.07						−6.22
	Orange Farm	57.25							−5.42	−3.10
	Randwater				47.19	−4.40	−0.67	+1.15	−27.53	−2.85
	Rosslyn			20.08	−0.75	−0.71				−0.15
Mpumalanga	Camden				53.95		−20.50		−3.41	−6.07
	Club			29.18	+14.16	+26.17		−6.38	−21.11	+0.02
	Embalenhle			37.08		+31.35			−4.38	+2.62
	Ermelo	40.50	+0.01	+15.93	−2.11	−37.80	+52.56	−10.68		−0.59
	Grootvlei		33.29		+2.36					+0.79
	Hendrina	38.23	−6.02	+11.23			−4.80	−29.07		−1.89
	Komati		64.56	+0.15		−7.33	+16.62	−15.79	+29.65	−0.06
	Middleburg	39.80	−20.34	+3.71	−19.66	−41.27	+50.82	−40.36		−3.84
	Phola				74.89	+1.90		−4.89	−6.57	−2.86
	Secunda	61.95	+1.09	−68.07	+122.87					−8.73
	Verykkop				24.39			−2.97	−21.07	−1.49
	Witbank	44.47			−1.12	−38.39	+119.33	−8.46		+1.58
Western Cape	Beliville			21.88	+10.84		+7.03	+0.05	+11.88	+1.65
	Foreshore		21.30	+3.40	−7.65		+15.22	−6.39	+15.28	+1.21
	George	21.95			−7.65					−1.68
	Goodwood		28.01	−8.44		+1.21	+12.64	+6.94		+0.84
	Stellenbosch			16.72	−0.09					−0.02
	Tableview		19.76	−8.11	−1.19					−0.91
	Wallacedene			16.91			+16.30	+31.45	+12.38	+4.05
KwaZulu-Natal	Brackenham		26.41	+13.66			+0.33	−6.90	+10.64	+0.45
	CBD		23.01	+13.91	+0.33		+5.00	−16.35	+6.32	+0.24
	Esikhaweni							27.22	−20.64	−5.62
	Gangles	34.61	+12.97		+7.07	+3.59				+2.88
	Ferndale	16.14	−19.44	−9.17						−2.16
							Legend			
							1st annual PM_10_ average			
							Percentage decrease in annual PM_10_			
							Percentage increase in annual PM_10_			
							No Data			

**Table 2 ijerph-18-13348-t002:** The distribution of daily PM_10_ concentration in µg/m^3^ by province and site type.

	Site Classifications											
	Industrial				Residential				Traffic			
Province	N	Median	25–75% percentile	Min–Max	N	Median	25–75% percentile	Min-Max	N	Median	25–75% percentile	Min–Max
Gauteng	5114	29	17.5–43.9	7.03–139	4020	58.5	41.2–144	20.9–344	731	53.9	38.8–74.6	23.5–152
Western Cape	4750	22.3	16.5–29.7	9.68–82.5	2193	25.4	18.4–34.8	10.9–93.4	2922	21.1	16.4–27.7	10.7–74.0
Mpumalanga	18264	37.1	22.3–59.2	8.78–228	2921	37.7	21.9–61.4	9.1–216	NA	NA	NA	NA
KwaZulu-Natal	NA	NA	NA	NA	5114	26	16.5–37.2	7.27–130	23.6	23.6	18.3–32.2	13.1–79.9

NA: Not available.

**Table 3 ijerph-18-13348-t003:** The percentage of PM_10_ (µg/m^3^) concentration exceeding daily standards by province for the years 2010–2017.

Province	WHO Standard		NAAQS Standard	
	Number of Days Exceeding Daily Limit	% of Days Exceeding Daily Limit ^a^	Number of Days Exceeding Daily Limit	% of Days Exceeding Daily Limit ^a^
Gauteng	3820/9865	38.7	1605/9865	16.3
Mpumalanga	7139/21185	33.7	3104/21185	14.7
KwaZulu-Natal	549/7671	7.2	108/7671	1.4
Western Cape	272/9865	2.8	8/9865	0.1

^a^ The percentage of days PM_10_ concentration daily limits were exceeded based on the number of days with PM_10_ data for 2010–2017 divided by the total number of days with valid PM_10_ data. WHO standard; World Health Organization 2021 daily standard of 45 µg/m^3^. NAAQS; South Africa’s National Air Quality 2006 daily standard of 75 µg/m^3^.

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
