# Peer review of "Spatial and Temporal Variations in PM10 Concentrations between 2010–2017 in South Africa"

_ijerph, 2021, doi:10.3390/ijerph182413348_

Round 1

Reviewer 1 Report

Spatial and temporal variations in PM10 concentrations between 2010-2017 in south Africa.

This study investigates spatio-temporal variations in PM10 in South Africa. If there has been no similar study in the area, this study would present valuable outcomes. However, to publish in the international journal, it should suggest more general implications and discussions. In my opinion, the study area is too specific to derive general implications.

My recommendations are as follows.

  1. The authors review more relevant literatures. And, they should compose literature review section. Current version does not include general section 2 (lit. review). Moreover, they only cited 20 references. I think there are lots of similar articles. I particularly recommend them to review papers focusing on countries with similar context with S. Africa.
  2. Based on the review, they may find common problems in those similar countries. Then, they can suggest more general implications for similar level of developing countries that does not have enough PM10 monitoring stations.
  3. In particular, I hope that Sections 4 and 5 explain the differences from the characteristics of other countries identified in previous studies. A detailed explanation of a specific country's situation is difficult to attract readers' interest.

Reviewer 2 Report

This is a very informative and valuable study on air quality, specifically PM10, in four provinces of SSA. Air quality in this area should be paid more attention but so far limited is available. The present study building on a previous study on imputing missing data in the studied area, focused on the pollution characteristics including monthly changes and sources.

The manuscript is generally clearly presented, but I think it needs a moderate revision before acceptance:

  1. all figures and table should be improved, and some may be moved to the appendix to increase the readability and visibility of the paper.
  2. the study aimed to analyze “source” – but it only discussed factors like buffers of land use categories, population density, road density, etc., Some are related to sources, but some not. Thus, I think it is more appropriate here to be an analysis on “influencing factors”, rather than sources which are usually power plants, vehicle emissions, residential combustions, etc., based on PM2.5 chemical component profiles and source apportionment methods like CMB and PMF.
  3. while ground observation data missed, it is helpful to look into and compare data from other technologies like satellite information and AOD-derived PM concentrations. Some products are available. The authors are suggested to compare the results with those, which can be better in capturing temporal and spatial changes of air pollutants in SSA.
  4. was the weekend effect significant? Statistic tests may be adopted in order to confirm the observation
  5. for the monthly and spatial differences in PM10, what about the influence of different meteorological conditions in data comparison?
  6. domestic burning explained high pollution levels in residential areas. Can you have more comments and information on residential fuel use and burning emissions from SSA on this issue? For example, how many population using dirty fuels in residential sector, and which types of fuels?

Round 2

Reviewer 1 Report

The authors properly revised the manuscript to reflect my comments. 

Reviewer 2 Report

the manuscript is clearly revised and organized